# *Bos taurus* and *Cervus elaphus* as Non-Seasonal/Seasonal Models for the Role of Melatonin Receptors in the Spermatozoon

**DOI:** 10.3390/ijms23116284

**Published:** 2022-06-03

**Authors:** Estela Fernández-Alegre, Estíbaliz Lacalle, Cristina Soriano-Úbeda, José Ramiro González-Montaña, Juan Carlos Domínguez, Adriana Casao, Felipe Martínez-Pastor

**Affiliations:** 1Institute of Animal Health and Cattle Development (INDEGSAL), University of León, 24071 Leon, Spain; efernandez@bianorbiotech.es (E.F.-A.); elacalle@bianorbiotech.es (E.L.); c.soriano.ubeda@unileon.es (C.S.-Ú.); jrgonm@unileon.es (J.R.G.-M.); jcdomt@unileon.es (J.C.D.); 2Bianor Biotech SL, 24071 Leon, Spain; 3Department of Molecular Biology (Cell Biology), University of León, 24071 Leon, Spain; 4Department of Medicine, Surgery and Veterinary Anatomy (Animal Medicine and Surgery), University of León, 24071 Leon, Spain; 5Department of Biochemistry and Molecular and Cell Biology, Institute of Environmental Sciences of Aragón, School of Veterinary Medicine, University of Zaragoza, 50013 Zaragoza, Spain; adriana@unizar.es

**Keywords:** melatonin, membrane receptor, GPCR, agonist, antagonist, spermatozoon, bull, deer, seasonality, reproduction

## Abstract

Melatonin is crucial in reproduction due its antioxidant, hormonal, and paracrine action. Melatonin membrane receptors (MT_1_/MT_2_) have been confirmed on spermatozoa from several species, but functionality studies are scarce. To clarify their role in ruminants as reproductive models, bull (*Bos taurus*, non-seasonal) and red deer (*Cervus elaphus*, highly seasonal) spermatozoa were analyzed after 4 h of incubation (38 °C, capacitating media) in 10 nM melatonin, MT_1_/MT_2_ agonists (phenylmelatonin and 8M-PDOT), and antagonists (luzindole and 4P-PDOT). Motility and functionality (flow cytometry: viability, intracellular calcium, capacitation status, reactive oxygen species (ROS) production, and acrosomal and mitochondrial status) were assessed. In bull, MT_1_ was related to sperm viability preservation, whereas MT_2_ could modulate cell functionality to prevent excess ROS produced by the mitochondria; this action could have a role in modulating sperm capacitation. Deer spermatozoa showed resistance to melatonin and receptor activation, possibly because the samples were of epididymal origin and collected at the breeding season’s peak, with high circulating melatonin. However, receptors could be involved in mitochondrial protection. Therefore, melatonin receptors are functional in the spermatozoa from bull and deer, with different activities. These species offer models differing from traditional laboratory experimental animals on the role of melatonin in sperm biology.

## 1. Introduction

Melatonin high-affinity receptors MT_1_ and MT_2_ have been detected in spermatozoa and reproductive tissues from different species [1,2,3,4]. These are G protein-coupled receptors (GPCR) capable of modulating various intracellular signaling pathways [5,6]. Their presence in the reproductive tract and gametes, together with the ability of these tissues for synthesizing melatonin [7], has driven many research lines and hypotheses about their purpose. Whereas melatonin is a potent antioxidant [8], the presence of these receptors indicates a paracrine activity, as demonstrated in the follicle [9,10] and male reproductive tract [1]. The role of the melatonin receptors on the spermatozoon is still not well known, but previous research suggests a relevant function, such as sperm maturation, capacitation or fertilization [11,12]. Melatonin has been found in the seminal plasma [2,13,14], possibly helping protect the cells from oxidative stress and modulating their physiology. This seminal plasma melatonin seems to be produced at least in part by tissues of the male reproductive system, mainly in the testicles and by the accessory glands, as demonstrated in rodents [5], sheep [7], bovines, and deer (unpublished).

Far from being a metabolically quiescent cell, the spermatozoon presents a dynamic metabolism [13]. From spermatogenesis, it undergoes many stages, such as maturation, capacitation, acrosomal reaction, and finally, the syngamy or fusion with the oocyte [14]. Once in the oviduct, the spermatozoon seems to be influenced by melatonin present in oviductal secretions but most relevantly in the follicular fluid [15]. Recent research has confirmed that melatonin could enhance sperm survival [16,17,18] and have a modulatory effect on sperm capacitation [18,19,20,21,22], a physiological process enabling the spermatozoon to fertilize the oocyte, eliciting its release from the oviductal epithelium, priming the acrosomal membrane for fusing with the plasma membrane (acrosomal reaction), and modifying the sperm membranes for fusion with the oolemma. These effects seem to be mediated by an action on signal transduction cascades by modifying the intracellular concentrations of second messengers and reactive oxygen species (ROS) [23,24]. Melatonin gradients in the oviduct from the follicular fluid, released after ovulation, could also contribute to the sperm’s guidance towards the oocyte by chemotactic action, possibly by regulating motility (in a process termed hyperactivation) [12,25].

Melatonin has been applied to reproductive technologies mainly for its antioxidant properties, but its effects at low concentrations through its membrane receptors could be similarly interesting [26]. Activity at nano or picomolar levels, which is compatible with its affinity to MT_1_ and MT_2_ [27,28], has been described in spermatozoa from many species. Research in the sheep model has provided relevant information. Thus, MT_1_ was found to have a variable distribution on the sperm head, depending on the spermatozoa and the individual male, whereas MT_2_ seems to be more restricted to the post-acrosomal and neck regions [3]. The confirmation of the presence of the receptors on spermatozoa and the activity of melatonin at low concentrations, which is compatible with receptors’ affinity [19], confirms that these receptors were functional. These studies also suggest that the activation of these receptors could contribute to preserving sperm viability while reducing capacitation [19,20]. Moreover, nanomolar melatonin decreased sperm cAMP intracellular concentration [20]. This activity resembles results in somatic cells, in which melatonin membrane receptors activate inhibitory G proteins, reducing cAMP levels and decreasing PKA activity [27,29]. In a previous report, our group found an important differential response of bull spermatozoa to a range of melatonin concentrations (1 pM to 1 µM) relative to motility, cell apoptosis, and other physiological parameters, which is compatible with receptor activation due to the low range of concentrations used [18].

Melatonin has also raised increased attention with respect to human fertility [30,31]. The possible association of melatonin with sperm physiology has opened a new field of research, and the use of this antioxidant and hormone for improving fertility problems is a fascinating prospect. In this regard, research on different animal models has provided valuable information. Pigs and ruminants offer an advantage over typical lab rodents due to a lack of ethical issues and sample availability (no need for animal sacrifice, availability from commercial stud centers), as well as quantity, with an acceptable homogeneity since males are selected for fertility and samples are routinely checked for high quality. Thus, our interest in the present study is advancing the research on the bovine model (*Bos taurus*) to confirm that melatonin modulates a key event in sperm physiology as the capacitation and, in some cases, preserves the cell viability and functionality. With our previous research suggesting that melatonin effects in sperm motility and physiology could be mediated by MT_1_/MT_2_ activation, we adopted a mechanistic strategy to define the role of these receptors by using specific agonists and antagonists while including melatonin to differentiate from its antioxidant effects (or a possible effect on other intracellular targets).

In addition, we have included samples from a wild ruminant, the red deer (*Cervus elaphus*), taking advantage of the sample availability and expanding the knowledge of the Artiodactyla group. The males of domestic ruminants have lost seasonality in part (sheep) or almost totally (many commercial cattle breeds). This is not the case for the red deer, with a dramatic seasonal variability in sexual activity and a total spermatogenic arrest in the non-breeding season [32]. This is especially relevant in studies with melatonin, since it is a critical player in the seasonal regulation of sexual activity, with direct and indirect implications for the reproductive organs [25,33]. Indeed, melatonin’s effects on human health, through alterations in circadian rhythms, for instance, have been proposed as one of the causes of reproductive dysfunctions and even other health issues [34,35], and this kind of model could be relevant for some lines of research [36]. This study could also shed some light on the role of melatonin receptors in the spermatozoon from a heavily seasonal species.

Therefore, our objective is to determine if melatonin receptors are responsible for the physiological modulation previously observed in vitro under a range of melatonin concentrations. We aim to discriminate which effects could be due to receptor activation or if other melatonin effects could have a role. These findings could help understand sperm physiology, move forward in solving some infertility problems, and enable the development of novel artificial reproductive techniques.

## 2. Results

### 2.1. Melatonin Receptor MT_2_ Contributes to Maintaining Bull Sperm Functionality

The analysis of flow cytometry data for bull sperm physiology (Figure 1) suggested that the direct activity of the MT_2_ receptor, rather than the overall action of melatonin, could modulate the viability and functionality of the cell. Compared to CTLm (Control with 10 nM melatonin), the specific MT_2_ antagonist 4P-PDOT produced a significant decrease in viability, in both PI^−^ and YO-PRO-1^−^ populations (Figure 1a,b, *p* < 0.05). It also produced a significant increase in the acrosomal (Figure 1d, *p* < 0.05), and reduced active mitochondria (Figure 1e, *p* < 0.05). The exception was the proportion of cells with high mitochondrial superoxide production (Figure 1h), in which CTL showed the highest levels, being significantly different from all treatments except for 4P-PDOT. This result suggests melatonin’s modulatory and possibly straightforward effect on the sperm mitochondria. However, there were no significant effects on the apoptotic sperm ratio (Figure 1c) in any of the treatments. Notice that the parameters presented acrosomal reaction (ratio) (Figure 1d), active mitochondria (Figure 1f) and ROS (Figure 1g,h) are expressed as the proportion (Figure 1d,g,h) or MFI (Figure 1g) within the viable population (PI^−^ or YO-PRO-1^−^).

Melatonin or its agonists (PHE, 8M-PDOT) or antagonists (LUZ, 4P-PDOT) did not produce any additional effects on the expected calcium uptake or [Ca^2+^]_i_ increase after the calcium ionophore challenge (*p* > 0.05) (Figure 2a–c). However, the lysophosphatidylcholine challenge (LPC; Figure 2d–f) elicited several differences in membrane disorder (Figure 2d), a proxy for sperm capacitation. This variable was only increased by phenylmelatonin (PHE) and 8M-PDOT treatments compared to the CTLm; however, the LPC challenge increased it significantly above the control (except for luzindole, LUZ; Figure 2e). Ratios were calculated as the value after the challenge with respect to the value of the untreated sample, to better estimate the amount of change. For these ratios, melatonin and 8M-PDOT showed significantly higher results than the control (Figure 2f). These results support a relevant role for MT_2_-activating pathways, increasing membrane fluidity and possibly enhancing a capacitation-like state, but not affecting [Ca^2+^]_i_.

### 2.2. Melatonin Receptors Modulate the Response of Bull Spermatozoa to Triggering the Acrosomal Reaction and Intracellular Calcium Distribution

The occurrence of the acrosomal reaction is shown in Figure 3. The differences in the untreated sample were minor (Figure 3a), with only a significant difference from the 10 nM melatonin CTLm to the 4P-PDOT treatment (tending to increase reacted acrosomes). This again suggests a role of MT_2_ in maintaining the acrosomal integrity, not compensated by the antioxidant effect of melatonin. The ionophore challenge’s induction of the acrosomal reaction did not change this pattern (Figure 3b). However, when analyzing the ratio among challenged/unchallenged (Figure 3c), the control showed a higher ratio than any of the agonists or antagonists. When LPC was used for the challenge, the increase in the acrosomal reaction was much larger than when using the ionophore (Figure 3d), but the differences between treatments were not evident. In this analysis, however, the ratio reflected the same trend as the ionophore treatment (Figure 3e). Therefore, both the receptors’ stimulation and melatonin (even at in presence of the antagonists) could downregulate the acrosomal reactivity when induced by these external agents.

**Figure 2 ijms-23-06284-f002:**
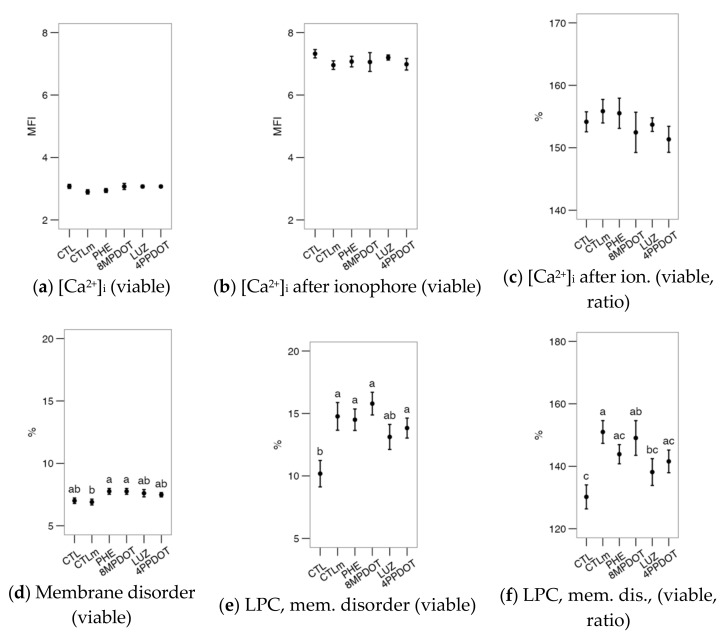
Effects on bull sperm intracellular calcium ([Ca^2+^]_i_) and membrane lipidic disorder (within the viable population) of melatonin receptor agonists (PHE: Phenylmelatonin; 8MPDOT: 8-Methoxy-2-propionamidotetralin) or antagonists with 10 nM melatonin (LUZ: Luzindole; 4PPDOT: cis-4-Phenyl-2-propionamidotetralin); CTL: Control; CTLm: Control with 10 nM melatonin. Plots show mean ± SEM for each treatment of seven males. (**a**) Intracellular calcium concentration ([Ca^2+^]_i,_ from mean fluorescence intensity of Fluo-4 in viable spermatozoa); (**b**) Same parameter after ionophore treatment and (**c**) as ratio of the values after (**b**) and before (**a**) the challenge; (**d**) Proportion of cells with increased membrane disorder in the viable population (PI^−^) as M540^+^ and (**e**) the same parameters after lysophosphatidylcholine (LPC) treatment, and (**f**) as ratio of the values after (**e**) and before (**d**) the challenge. Treatments not sharing the same letters differ significantly (*p* < 0.05).

When analyzing the Ca^2+^ patterns with the CTC stain (Figure 4), it was found that PHE and 4P-PDOT, respectively, decreased (*p* < 0.05) the F (uncapacitated) and the B (Figure 4b, capacitated) patterns compared to the control. Differences were observed between treatments, with a slight increase in the B pattern with PHE compared to antagonist treatments (LUZ and 4P-PDOT) but non-significant compared to the control and CTLm. There were no additional effects on the AR pattern of melatonin or its agonists or antagonists. Nevertheless, the combined results for F and B suggest a combined functionality of MT_1_ and MT_2_ on the Ca^2+^ distribution during the capacitation process.

### 2.3. The Activation of Melatonin Receptors Improves Sperm Motility

Previously [18], we found that melatonin showed little effect on bull sperm motility and even decreased it at 100 nM. Melatonin agonists (PHE and 8M-PDOT) significantly increased total sperm motility (Figure 5a) concerning the CTL and CTLm, indicating an activator role for at least the MT_2_ receptor (since 8M-PDOT, which is much more specific for MT_2_). Nevertheless, compared to the control, the presence of melatonin (CTLm, LUZ and 4P-PDOT) increased the average values for total motility (Figure 5a), but only significantly (*p* < 0.05) for 4P-PDOT. This shows that melatonin could have some effect at 10 nM independently of melatonin receptors. Interestingly, melatonin decreased the proportion of progressive spermatozoa in comparison to CTL (Figure 5b; significantly for CTLm).

**Figure 4 ijms-23-06284-f004:**
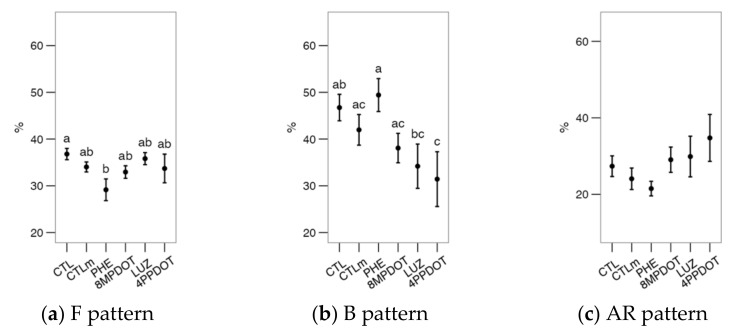
Effects on bull sperm CTC patterns (Ca^2+^ distribution from CTC stain) of melatonin receptors agonists (PHE: Phenylmelatonin; 8MPDOT: 8-Methoxy-2-propionamidotetralin) or antagonists with 10 nM melatonin (LUZ: Luzindole; 4PPDOT: cis-4-Phenyl-2-propionamidotetralin); CTL: Control; CTLm: Control with 10 nM melatonin. Plots show mean ± SEM for each treatment of seven males. (**a**) % of spermatozoa showing F pattern (non-capacitated spermatozoa showing CTC fluorescence over the head); (**b**) % of spermatozoa showing B pattern (capacitated spermatozoa showing CTC fluorescence on the acrosomal-equatorial region); (**c**) % of spermatozoa showing AR pattern (acrosome-reacted spermatozoa showing CTC fluorescence as a band on the equatorial region). Treatments not sharing the same letters differ significantly (*p* < 0.05).

The cluster analysis yielded three subpopulations from the bull sperm motility data (Table 1), named “slow” (slow and erratic swimming), “fast” (faster and most linear swimming), and “active” (the fastest swimming and with more lateral movement of the head by ALH while maintaining LIN). The “active” subpopulation was predominant in this experiment, although their relative proportions varied among the experimental groups (Figure 5c–e). Whereas we obtained no significant effects for the proportions either of the “slow” nor “fast” subpopulations (Figure 5c,d), the melatonin agonists (PHE and 8M-PDOT) significantly increased the proportion of “active” spermatozoa concerning the control (Figure 5e). Another relevant finding is that LUZ caused a drop in the proportion of these “active” spermatozoa compared to PHE, 8M-PDOT, and 4P-PDOT (thus, possibly mediated by MT_1_ blocking). The lack of significance, when compared with CTLm, supports a receptor-mediated mechanism in this case.

### 2.4. Melatonin and the Receptors Agonists/Antagonists Showed Few Effects on Deer Spermatozoa

After the incubation, the evaluation of the sperm samples showed no significant differences among treatments for most parameters: viability as PI^−^ and YO-PRO-1^−^, the proportion of apoptotic cells and reacted acrosomes in the viable population, the concentrations of cytoplasmic ROS and mitochondrial O_2_^•−^, acrosomal reaction and the Ca^2+^ patterns by CTC stain (Figure 6, Figure 7, Figure 8 and Figure 9). Only the proportion of cells with higher mitochondrial activity varied significantly (Figure 6e,f, as proportion in the viable population). The control with 10 nM melatonin (CTLm) showed a significantly higher average value of active mitochondria than 4P-PDOT (Figure 6e), while considering only the viable cells (Figure 6f), the control was significantly lower than the remaining treatments. These results suggest a mitoprotective effect of melatonin, which may be due to direct activity in the mitochondria but also dependent on MT_2_ activation, which is similar to the findings in the bull.

Interestingly, [Ca^2+^]_i_ was significantly higher in the controls than in the 4P-PDOT treatment (Figure 7a). Whereas there were no significant effects after the calcimycin challenge (Figure 7b), the ratio between the challenged cells and the basal analysis showed a relative response of the cells treated with the antagonists LUZ and 4P-PDOT (Figure 7c), overall suggesting a role of the receptor’s inhibition regarding the control of intracellular calcium. However, the proportion of cells with membrane disorder was kept low in all treatments, including CTLm, compared to the control (Figure 7d). The LPC challenge induced a general increase in the proportion of these cells (*p* < 0.05 in a general comparison of unchallenged and challenged samples) and showed a higher response in the 8M-PDOT and the antagonists (LUZ and 4P-PDOT) in comparison with CTL; Figure 7e). The ratio (Figure 7f) helped interpret these contradictory results by showing that the only treatment significantly different from the control was 8M-PDOT and that MT_2_ could affect pathways related to the membrane response to the challenge.

The acrosomal reaction was unaffected by any treatment (Figure 8a), with no apparent differential responses after the calcimycin or LPC challenges (Figure 8b,e). There were no significant effects of the treatments on the CTC pattern (Figure 9) or motility (Figure 10). Considering the sperm motility cluster analysis, it yielded three subpopulations as with the bull sperm analysis (Table 2). Similar to that case, the “active” subpopulation was predominant. However, the treatments did not contribute significantly to the variability within subpopulations (Figure 10c,d).

## 3. Discussion

Melatonin has been applied in reproductive medicine and specifically in assisted reproductive techniques for its antioxidant and cytoprotective properties [32,37,38]. When applied to sperm preservation, the concentrations are usually above the physiological ones found in the reproductive fluids (the nano- to the picomolar range) [2,39]. A direct antioxidant activity does not seem relevant at these concentrations, likely modulating intracellular pathways [40] and binding to the G protein-coupled receptors (GPCR) MT_1_ and MT_2_. Currently, the modulation of the pathways related to antioxidant activity has gained strong support from studies mainly in the sheep model [22,25]. Considering that these receptors have been detected in different mammal species [2], these mechanisms could be widespread in spermatozoa. In a previous study, low melatonin concentrations elicited physiological responses (such as decreased sperm motility or cell apoptosis and increased mitochondrial activity) in bull spermatozoa compatible with receptor binding [18], with some evidence for deer spermatozoa [37].

The most relevant result from our study with the bovine model (bull spermatozoon) is the relevance of MT_2_ as a putative modulator of sperm physiology. The activity of this receptor seems to be associated with an antiapoptotic effect, since the specific antagonist 4P-PDOT decreased viable and non-apoptotic spermatozoa, which aligns with previous reports on the sheep model [22]. A previous study in human spermatozoa suggested a role for MT_1_ [16] in maintaining sperm viability through the activation of ERK-related pathways. Nevertheless, melatonin was used at 1 mM in that study, and therefore a confounding antioxidant effect could not be discarded.

Our previous study suggested a role of melatonin at low concentrations in promoting sperm viability, which is compatible with receptor activation [18]. Indeed, the specific blocking of MT_2_ by 4P-PDOT could disrupt cell homeostasis or the redox balance, considering the increase of cytoplasmic ROS. These events could also be related to the higher susceptibility of spermatozoa to spontaneously undergo acrosomal reaction (a complex event including the fusion of plasma and acrosomal membranes [38]).

Melatonin has been praised for its beneficial effects on mitochondria, and in fact, this molecule can accumulate in the mitochondrial matrix, maximizing its antioxidant and modulatory activity [40,41]. In addition to its antioxidant action, melatonin increases the activity of mitochondrial complexes 1, 3, and 4, maintaining ATP production, and reduces the duration of transition pore opening, balancing the membrane potential [42]. In the present study, melatonin and its receptors showed some mitoprotective activity, evident in the reduced superoxide radical production in this organelle. In both the sheep [40] and bovine [18] models, melatonin reduced the sperm mitochondrial superoxide production irrespective of concentration while not affecting (or only slightly affecting) cytoplasmic ROS. Melatonin can decrease general ROS levels in the ram [20], red deer [43], and bull [44] spermatozoon, but at higher concentrations than those considered in this study and therefore exerting a direct (not receptor-mediated) antioxidant effect. Our results support the role of membrane receptors, and more likely MT_2_, for regulating the redox balance mediated by an effect on mitochondria.

Nevertheless, the mechanism for transducing the signal to the mitochondria is still to be clarified. In fact, melatonin could also affect the mitochondria directly (and specifically, O_2_^−•^ modulation), and some studies have proposed a mechanism for its transport within this organelle [45]. Moreover, high-affinity intracellular targets have been suggested, and indeed, MT_1_ has been identified in the mitochondrial outer membrane [46].

Furthermore, our results suggest a role of the melatonin receptors in the regulation of sperm capacitation, a critical event in the life of this cell type and necessary for achieving fertilization capacity. In our previous study in bull spermatozoa [18], melatonin seemed to promote capacitation-like processes (however, only in a non-capacitating medium), and our results agree with a similar effect in the hamster [12] and in the sheep models [19,20]. In this regards, Ca^2+^ is an essential agent in the first phases of capacitation. Although [Ca^2+^]_i_ was not affected among treatments, the intracellular distribution of this ion was modified, indicating an enhancement in capacitation in part of the sperm population when the receptors (possibly MT_1_) were stimulated. Our results agree with studies in ram spermatozoa using the CTC technique for analyzing the Ca^2+^ patterns, which have provided evidence for the role of melatonin in the modulation of this process [19,20]. Melatonin at micromolar concentrations decreased markers of capacitation, whereas it increased short-term capacitation and in vitro fertility at picomolar concentrations. Since the antioxidant effects of melatonin are negligible at nano- to picomolar scales, the effects of low melatonin concentrations on capacitation (at least in ruminants) are likely mediated by the high-affinity membrane receptors. At high concentrations (µM and mM), these antioxidant effects might inhibit capacitation by removing free radicals (critical mediators in this process).

Moreover, melatonin could also modulate cAMP levels, as shown in ram spermatozoa [20]. This might explain the apparent lack of effect of receptor agonists and antagonists on [Ca^2+^]_i_, and therefore techniques for assessing [cAMP], should be included to achieve a full understanding of the role of melatonin receptors.

Some of the events related to capacitation include the change in the sperm motility to a hyperactivated state (vigorous, large-bend flagellar movements), mainly due to [Ca^2+^]_i_ increase after the opening of specific channels in the plasma membrane, such as CatSper [47]. Few studies have dealt with this issue regarding the effect of melatonin receptors. An early study on hamster suggested that melatonin increased sperm hyperactivation [12], but this effect was not observed in ruminants, preserving total motility and promoting progressivity [20,48]. Nevertheless, melatonin did not affect sperm motility in bull [18], and even depressed it at 100 pM in capacitating conditions. In the present experiment, we used 10 nM melatonin, reporting no change in total sperm motility while losing sperm progressivity. Interestingly, the stimulation of the receptors, putatively MT_2_, increased total motility while maintaining progressivity. Therefore, melatonin could modulate motility in this model by affecting the cell at different levels, possibly competing for receptor-mediated and intracellular targets. Interestingly, the “active sperm” subpopulation was kept at the same level as the control by luzindole, while being significantly increased by the specific MT_2_ inhibitor 4P-PDOT, suggesting a role of MT_1_ promoting this kind of motility. Specific studies on receptor dynamics, interactions, and modulation of intracellular pathways might help to elucidate whether our results and those in spermatozoa from other species [16,17,18,19,20] can be explained by these cellular processes.

Our results in the deer model showed a lower response to the different treatments compared to the bull model. A similar effect was observed when exposing deer spermatozoa to a range of melatonin concentrations [37]. Whereas we cannot discard a difference between species, the most likely explanation is that epidydimal spermatozoa could be refractory to melatonin and receptors’ agonists/antagonists. Previously, we noticed important differences between epididymal and ejaculated spermatozoa in deer and other ruminants [49,50,51], apparently showing better resilience to adverse conditions in the former because of a lack of contact with seminal plasma. The epididymal spermatozoa are mature in the cauda epididymis and able to fertilize if diluted and stored in appropriate media. However, they are quiescent in the epididymal environment, whose osmolality, pH, and overall composition differs from other reproductive fluids, and where they are tightly packed at a very high density [52]. In fact, the dilution and contact with seminal plasma, even if brief (as is typical in ruminants), produces a wide range of changes and the activation or modulation of intracellular pathways, together with the modification of the plasma membrane and a wide range of peripheral proteins [53].

Additionally, the samples used in this study were collected around the peak of the breeding season, after the highest levels of blood plasma melatonin. In the ram model (mildly seasonal compared to the deer), levels in seminal plasma are the highest during autumn [39], with similar concentrations in the cauda epididymis [7]. A hypothesis to test in future studies is a possible desensitization effect of previous exposure to melatonin. In a previous study, 1 mM and higher concentrations showed a protective effect against oxidative stress in deer spermatozoa [43], but other physiological responses were subtler, hinting at the resilience of epididymal spermatozoa to respond to melatonin. Nevertheless, our results have shown that melatonin has a mitoprotective activity in this species, mainly by intracellular activity, since the control with melatonin achieved the highest results, and both 8M-PDOT and 4P-PDOT reduced the proportion of spermatozoa with active mitochondria (viable and total populations, respectively). Future studies could test ejaculated spermatozoa or apply heterologous seminal plasma to epididymal sperm, which might prime these cells for responding to the melatonin stimuli [54].

The role of melatonin receptors in sperm physiology is a fascinating topic, but we are far from grasping the complete picture. Whereas there are similar results in different cell types, it is clear that the effects and regulations are very diverse, and the spermatozoon stands out as a highly peculiar cell. Our results for the bull and deer spermatozoa suggest a great contrast in this cell type regarding its response to melatonin depending on the exposure to seminal plasma (epidydimal vs. semen/ejaculate environments). Melatonin receptors can form homo- or heterodimers [55], and they have been found in specific membrane microdomains. The melatonin receptors’ distribution on the sperm surface has been primarily studied in the sheep model [3,22], with results confirmed in other species [2], suggesting that homodimers could be more frequent, but heterodimers could occur in some membrane domains. These domains change with capacitation (sheep [22] and unpublished results for bull and deer) and possibly with exposure to seminal plasma and other reproductive fluids (of particular interest for the case of epididymal spermatozoa). Receptor functionality could vary depending on membrane distribution because of homo- or heterodimerization (which showed different activity in other cell types [56]) and because they could affect specific sperm compartments. A deep understanding of receptor dynamics (including possible internalization [22]) should complement and enable an understanding of how melatonin influences sperm physiology.

## 4. Materials and Methods

### 4.1. Reagents, Animals, and Sample Collection and Preparation

Reagents were purchased from Merck KGaA (Darmstadt, Germany). Fluorescence probes were purchased from ThermoFisher Scientific (Waltham, MA, USA). Flow cytometry solutions were purchased from Beckman Coulter (Brea, CA, USA). Melatonin was purchased from Merck KGaA (Sigma-Aldrich M5250); the receptor agonists and antagonists were purchased from Tocris Bioscience (Bristol, UK): phenylmelatonin (0680), 8M-PDOT (1035), luzindole (0877), and 4P-PDOT (1034).

Bull samples (*Bos taurus*) were donated by CENSYRA (Center for Animal Selection and Breeding, Junta de Castilla y León, León, Spain). The staff obtained the ejaculates following routine procedures and as part of the center’s activity producing frozen semen doses for cryobanking and artificial insemination. After collecting the ejaculates (by artificial vagina [18,57]), the staff separated aliquots and handed them to the researchers and then transported them to the laboratory at 30 °C (less than 1 h from ejaculation). Seven samples (n = 7) were used in the study, from two Limousine and five Holstein bulls (12–24 months of age).

Immediately after arrival to the laboratory, the semen was washed to remove the seminal plasma. The sample was layered on 7.5 mL of a sucrose cushion (10 mM NaCl, 2.5 mM KCl, 20 mM HEPES, 222 mM sucrose, 5 mM glucose and 0.1% PVP; pH 7.3) and centrifuged (200× *g* 5 min, 900× *g* 10 min), and the pellet was adjusted to 50 × 10^6^ mL^−1^ in TALP-HEPES medium (100 mM NaCl, 3.1 mM KCl, 25 mM NaHCO_3_, 21.6 mM Na lactate, 10 mM HEPES, 5 mM glucose, 3 mM CaCl_2_, 1 mM Na pyruvate, 0.3 mM NaH_2_PO_4_, 0.4 mM MgCl_2_, 2 U/mL gentamicin, 0.5% phenol red, and 0.5% BSA; pH 7.3).

The red deer samples (*Cervus elaphus hispanicus*) were obtained from the Picos de Europa Hunting Reserve (Riaño, León, Spain) during October (around the peak of the breeding season). Immediately after the animals died, the wardens of the hunting reserve harvested testicles inside the scrotum, packed them in plastic sample storage bags, and handed them to the researchers. The testicles of five different males (n = 5) were transported to our laboratory in Styrofoam boxes with ice packs to maintain inside temperature and processed the following day (24–36 h post mortem). The age of the hunted stags was estimated at around 3 to 6 years old (both shipping and stag age are not considered factors for sperm quality [58,59]). As described previously, samples were obtained by dissecting and cutting the cauda epididymis [59,60]. The sperm mass was adjusted to 50 × 10^6^ mL^−1^ in cold TALP-HEPES medium and left to warm to 30 °C.

### 4.2. Experimental Design

The samples were assessed 15 min after extending in TALP-HEPES medium and warming to 38 °C. Then, sodium heparin was added to 2 U/mL (for promoting capacitation), and the samples were split into six tubes. The influence of melatonin receptors on cellular physiology was tested by adding to different tubes the MT_1_/MT_2_ agonist phenylmelatonin (10 nM, PHE), the MT_2_ agonist 8-methoxy-2-propionamidotetralin (50 nM, 8M-PDOT), the MT_1_/MT_2_ antagonist luzindole (100 nM, LUZ), and the MT_2_ antagonist cis-4-phenyl-2-propionamidotetralin (100 nM, 4P-PDOT); both tubes with antagonists also received melatonin at 10 nM, in order to determine melatonin effects not mediated by MT_1_/MT_2_. The remaining two tubes were used as controls. One received melatonin at 10 nM (CTLm) (the same concentration as antagonist tubes), and the other received no drugs (CTL). All drugs (agonists, antagonists, and melatonin) were prepared in DMSO, and all tubes (including the controls) received the same amount of vehicle as 0.5% DMSO. The drug’s concentrations were determined according to previous studies on spermatozoa, based on each receptor’s agonist or antagonist effect [12,22].

Duplicate analyses were carried out in the original sample (after extension in TALP-HEPES) and after incubating the tubes for 4 h at 38 °C, 5% CO_2_. Results from the initial assessment are shown in the Appendix A.

### 4.3. Flow Cytometry Analysis of Sperm Physiology

The response of the cells to the treatments was tested with fluorescence probes and flow cytometry as described previously [61,62,63]: Hoechst 33342 (H342, for debris discrimination, specifically labeling nuclei) at 5 µM; YO-PRO-1 (YP1, apoptotic changes detection) at 100 nM; Fluo-4 (intracellular Ca^2+^ concentration assessment) at 100 nM; CM-H_2_DCFDA (CFDA, cytoplasmic ROS) at 5 µM; Merocyanine 540 (M540, plasmalemma changes related to capacitation) at 2 µM; propidium iodide (PI, viability) at 1 µM; PNA-Alexa Fluor 647 (PNA, peanut agglutinin, acrosomal integrity) at 1 µg/mL; MitoTracker deep red (MT, mitochondrial status) at 100 nM; and MitoSOX (MSX, mitochondrial superoxide assessment) at 1 µM. The probes were combined in TALP-HEPES as H342/Fluo-4/M540/PI/PNA, H342/CFDA/PI, and H342/YP/MSX/MT.

Spermatozoa were adjusted at 10^6^ mL^−1^ in each probe combination. After 15 min in the dark at 37 °C, the tubes were analyzed with a CyAn ADP flow cytometer (Beckman Coulter). The cytometer included three diode lasers (405 nm, 488 nm, and 635 nm). The fluorescence was detected with photodetectors provided with specific filters: for the 405 nm line, we used filters 450/50 (H342); in the 488 nm line, filters 530/40 (YO-PRO-1, Fluo-4, H_2_DCFDA), 575/25 (M540) and 613/20 (PI, MitoSOX); and in the 633 nm line, filter 665/20 (PNA-Alexa Fluor 647, MitoTracker deep red). The acquisition was controlled with the Summit V4.3.02 software. Cytometry data were saved as FCS v.3 files and analyzed with the Weasel v. 3.5 software (Frank Battye, Melbourne, Australia). The cells were plotted as FSC/SSC (forward/side scatter) and H342/SSC, gating the events corresponding to spermatozoa for extracting the parameters. Viability was determined as YP^−^ (viable non-apoptotic) and PI^−^ events; PNA^+^ events were considered as spermatozoa with reacted acrosomes; the ratio of reacted acrosomes in viable spermatozoa was calculated from the PNA^+^ events in the PI^−^ population; the apoptosis ratio was calculated from the YP^−^ and PI^−^ events; the capacitation ratio was estimated from M540^+^ in the PI^−^ population; the mitochondrial activity as MT^+^/YP^−^ spermatozoa; the mitochondrial superoxide production was expressed as the ratio of viable cells from MSX^+^ events in the YP^−^ population. These parameters were expressed as percentages (%). The mean fluorescence intensity (MFI) of Fluo-4 and CFDA in viable cells (PI^−^) was used for comparing the intracellular Ca^2+^ concentration ([Ca^2+^]_i_) and cytoplasmic ROS presence, respectively.

### 4.4. Ionophore and Lysophosphatidylcholine Challenges

After 4 h incubation under capacitating conditions described in “Experimental design,” we tested the susceptibility of spermatozoa to increase intracellular calcium and the pharmacologically induced acrosome reaction with a Ca^2+^ ionophore (calcimycin) and lysophosphatidylcholine (LPC). Calcimycin (3 µM)and LPC (0.3 mg/mL) were added to separate tubes with H342/Fluo-4/M540/PI/PNA, and they were assessed after 10 min at 37 °C by flow cytometry [18,22]. For calcimicyn, [Ca^2+^]_i_ increase was assessed by Fluo-4, and for LPC, the membrane instability was assessed by M540, as described above. In both cases, the acrosomal reaction was assessed by PNA. In order to better assess each parameter change, we calculated the ratio by dividing the value after the challenge by the value of the unchallenged sample (naive, as reference).

### 4.5. Chlortetracycline (CTC) Stain for Assessing [Ca^2+^]_i_ Patterns

Chlortetracycline (CTC) is an antibiotic with a high affinity for Ca^2+^ and the ability to emit a green-yellow fluorescence upon binding this ion. It can be used to track [Ca^2+^] increases in the sperm cytoplasm [64]. Being a highly compartmentalized cell, the spermatozoon presents different fluorescence patterns according to its capacitation status (F: non-capacitated, mostly homogeneous fluorescence over the head; B: Capacitated, fluorescence on the acrosomal-equatorial part of the head, with a dark post-acrosomal region; AR: Acrosome-reacted, with loss of fluorescence in the anterior part, frequently with a brighter equatorial band). An 18-µL sample aliquot was mixed with 20 µL of CTC solution (750 µM CTC in 20 mM Tris, 130 mM NaCl and 5 mM cysteine, pH 7.8) and 5 µL of 12% formaldehyde solution (0.5 Tris-HCl, pH 7.8). After 10 min in the dark at 37 °C, the sample was mounted on a glass slide with a drop of 0.22 M DABCO (triethylenediamine) in glycerol:PBS (3:1), sealed with enamel, and stored at 4 °C in the dark. The slides were examined with an epifluorescent microscope at ×40 (Nikon E600, Nikon, Tokyo, Japan) and using a V-2A filter. At least 150 spermatozoa were assessed, discriminating between the F, B, and AR patterns.

### 4.6. Sperm Motility Analysis

An aliquot was extended in TALP-HEPES at 10^−8^ mL^−1^ and prepared in a modified Makler chamber (20 µm depth) on a warmed stage at 37 °C. The computer-assisted sperm analysis system consisted of a Nikon E600 microscope (Nikon, Tokyo, Japan) with ×10 negative phase contrast optics and a Basler A312fs digital camera (Basler Vision Technologies, Ahrensburg, Germany) working at 53 fps. The recorded videos were processed with the ISAS software (Proiser, Valencia, Spain), which provided motility parameters for each cell. At least 200 motile cells in 5 different fields were captured in each sample. Motility parameters [65] were VCL (curvilinear velocity), VSL (straight path velocity), VAP (average path velocity according to the average smoothed path; µm/s), LIN (linearity), STR (straightness), WOB (wobble), ALH (amplitude of the lateral displacement of the sperm head), BCF (frequency of the flagellar beat), DNC (sperm dance), and DNCm (sperm mean dance). The data analysis yielded total (MOT, as VCL > 10 µm/s) and progressive motility (PROG, as VCL > 25 µm/s and STR > 80%). Raw data were processed with R scripts [66], performing a cluster analysis (two-step procedure applying the AGNES clustering algorithm) [67]. This cluster analysis provided 3 subpopulations.

### 4.7. Statistical Analysis

Data were analyzed in the R statistical environment [66]. Since sperm samples were obtained from different males with no pooling (seven for bull; five for deer), multilevel modeling was used to account for between-male variability, employing linear mixed-effects models [68] with the treatment as a fixed effect and the male as a random effect. Results were expressed as mean ± SEM of each response parameter.

## 5. Conclusions

The membrane receptors MT_1_ and MT_2_ modulate sperm physiology in bull-ejaculated spermatozoa. MT_2_ could be especially relevant considering its mitoprotective activity and the modulation of intracellular redox balance. Melatonin plays an essential role in some critical events for sperm functionality, such as capacitation, resembling observations in the sheep model but with several specific differences. The experiment with deer epididymal spermatozoa suggested a possible resiliency to melatonin receptor activation or inactivation. These results should be complemented by future research on receptor dynamics and modulation of intracellular pathways upon melatonin exposure. These different models can provide valuable information for better understanding the role of this critical molecule on fertility and improving reproductive techniques in human medicine and animal production.

## Figures and Tables

**Figure 1 ijms-23-06284-f001:**
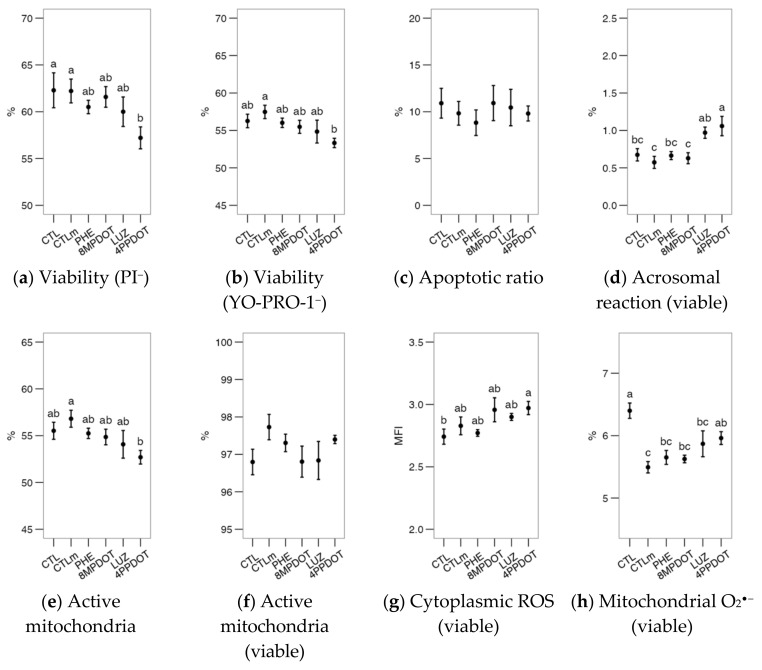
Effects on bull sperm physiological parameters of melatonin receptors agonists (PHE: Phenylmelatonin; 8MPDOT: 8-Methoxy-2-propionamidotetralin) or antagonists with 10 nM melatonin (LUZ: Luzindole; 4PPDOT: cis-4-Phenyl-2-propionamidotetralin); CTL: Control; CTLm: Control with 10 nM melatonin. Plots show mean ± SEM for each treatment of seven males. (**a**) Viable spermatozoa as maintaining membrane continuity (PI^−^); (**b**) Viable, non-apoptotic spermatozoa (as YO-PRO-1^−^); (**c**) Apoptotic spermatozoa (YO-PRO-1^+^) as ratio of PI^−^; (**d**) Acrosome-reacted spermatozoa (PNA^+^); (**e**) Spermatozoa with active mitochondria (MitoTracker^+^); (**f**) Spermatozoa with active mitochondria as a proportion of the viable population; (**g**) Cytoplasmic ROS production (H_2_DCFDA^+^) of viable spermatozoa; (**h**) Spermatozoa with high mitochondrial superoxide production (MitoSOX^+^) as a proportion of the viable population. Treatments not sharing the same letters differ significantly (*p* < 0.05).

**Figure 3 ijms-23-06284-f003:**
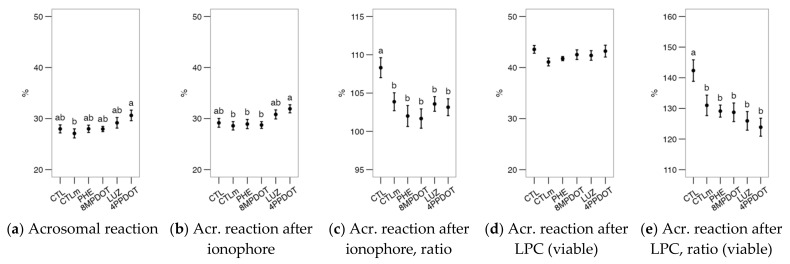
Effects on bull sperm acrosomal reaction (within the viable population) of melatonin receptors agonists (PHE: Phenylmelatonin; 8MPDOT: 8-Methoxy-2-propionamidotetralin) or antagonists with 10 nM melatonin (LUZ: Luzindole; 4PPDOT: cis-4-Phenyl-2-propionamidotetralin); CTL: Control; CTLm: Control with 10 nM melatonin. Plots show mean ± SEM for each treatment of seven males. (**a**) Acrosomal reaction (PNA^+^ cells) in neat, (**b**) after ionophore treatment, and (**c**) as a ratio of the values after (**b**) and before (**a**) the challenge; and (**d**) after lysophosphatidylcholine (LPC) treatment, and (**e**) as a ratio of the values after (**d**) and before (**a**) the challenge. Treatments not sharing the same letters differ significantly (*p* < 0.05).

**Figure 5 ijms-23-06284-f005:**
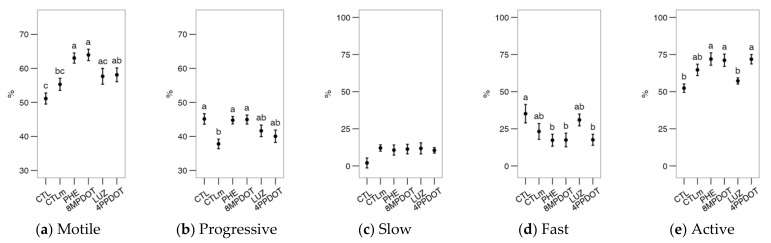
Effects on bull sperm motility of melatonin receptors agonists (PHE: Phenylmelatonin; 8MPDOT: 8-Methoxy-2-propionamidotetralin) or antagonists with 10 nM melatonin (LUZ: Luzindole; 4PPDOT: cis-4-Phenyl-2-propionamidotetralin); CTL: Control; CTLm: Control with 10 nM melatonin. Plots show mean ± SEM for each treatment of seven males. (**a**) Percentage of motile spermatozoa; (**b**) Percentage of progressive spermatozoa; (**c**) “Slow” spermatozoa, low velocity and linearity; (**d**) “Fast” spermatozoa, high velocity and linearity; (**e**) “Active” spermatozoa, highest velocity and high lateral movement. Treatments not sharing the same letters differ significantly (*p* < 0.05).

**Figure 6 ijms-23-06284-f006:**
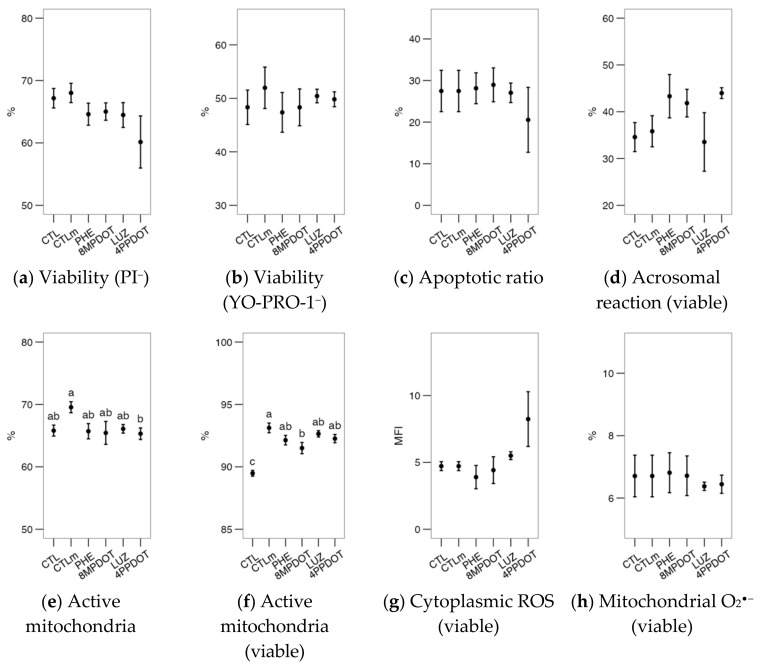
Effects on red deer epididymal sperm physiological parameters of melatonin receptor agonists (PHE: Phenylmelatonin; 8MPDOT: 8-Methoxy-2-propionamidotetralin) or antagonists with 10 nM melatonin (LUZ: Luzindole; 4PPDOT: cis-4-Phenyl-2-propionamidotetralin); CTL: Control; CTLm: Control with 10 nM melatonin. Plots show mean ± SEM for each treatment of five males. (**a**) Viable spermatozoa as maintaining membrane continuity (PI^−^); (**b**) Viable, non-apoptotic spermatozoa (as YO-PRO-1^−^); (**c**) Apoptotic spermatozoa (YO-PRO-1^+^) as ratio of PI^−^; (**d**) Acrosome-reacted spermatozoa (PNA^+^); (**e**) Spermatozoa with active mitochondria (MitoTracker^+^); (**f**) Spermatozoa with active mitochondria as a proportion of the viable population; (**g**) Cytoplasmic ROS production (H_2_DCFDA^+^) of viable spermatozoa; (**h**) Spermatozoa with high mitochondrial superoxide production (MitoSOX^+^) as a ratio of viable spermatozoa. Treatments not sharing the same letters differ significantly (*p* < 0.05).

**Figure 7 ijms-23-06284-f007:**
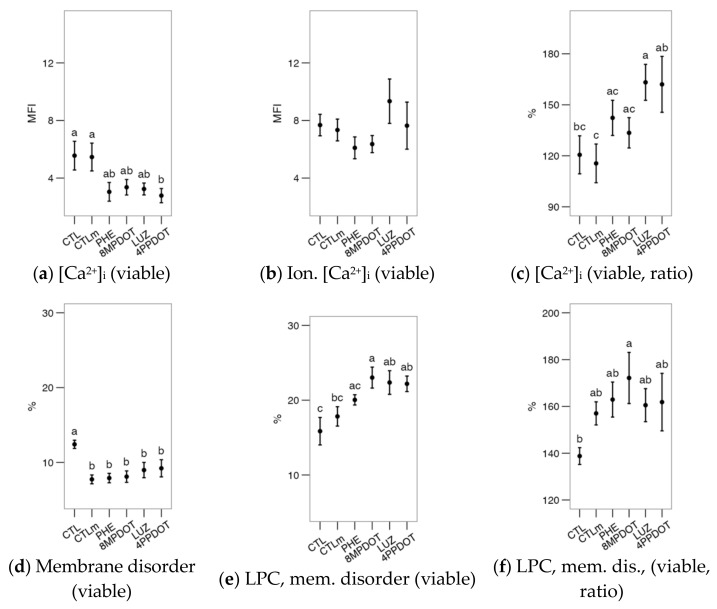
Effects on red deer epididymal sperm intracellular calcium ([Ca^2+^]_i_) and membrane lipidic disorder (within the viable population) of melatonin receptors agonists (PHE: Phenylmelatonin; 8MPDOT: 8-Methoxy-2-propionamidotetralin) or antagonists with 10 nM melatonin (LUZ: Luzindole; 4PPDOT: cis-4-Phenyl-2-propionamidotetralin); CTL: Control; CTLm: Control with 10 nM melatonin. Plots show mean ± SEM for each treatment of five different males. (**a**) Intracellular calcium concentration ([Ca^2+^]_i_, from mean fluorescence intensity of Fluo-4 in viable spermatozoa); (**b**) Same parameter after ionophore treatment and (**c**) as ratio of the values after (**b**) and before (**a**) the challenge; (**d**) Proportion of cells with increased membrane disorder in the viable population (PI^−^) as M540^+^ and (**e**) the same parameters after lysophosphatidylcholine (LPC) treatment and (**f**) as ratio of the values after (**e**) and before (**d**) the challenge. Treatments not sharing the same letters differ significantly (*p* < 0.05).

**Figure 8 ijms-23-06284-f008:**
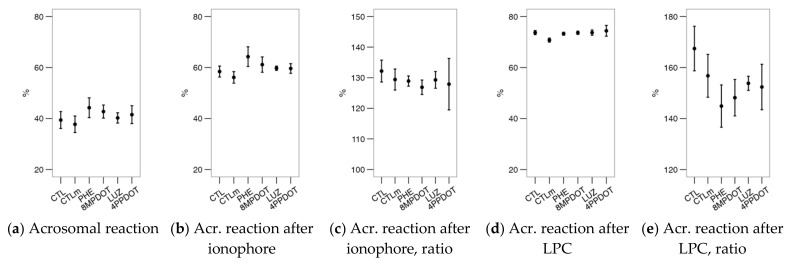
Effects on red deer epididymal sperm acrosomal reaction (within the viable population) of melatonin receptor agonists (PHE: Phenylmelatonin; 8MPDOT: 8-Methoxy-2-propionamidotetralin) or antagonists with 10 nM melatonin (LUZ: Luzindole; 4PPDOT: cis-4-Phenyl-2-propionamidotetralin); CTL: Control; CTLm: Control with 10 nM melatonin. Plots show mean ± SEM for each treatment of five males. (**a**) Acrosomal reaction (PNA^−^ cells) as the ratio of the viable sperm population (PI^−^), (**b**) after ionophore treatment and (**c**) as a ratio of the values after (**b**) and before (**a**) the challenge, and (**d**) after lysophosphatidylcholine (LPC) treatment and (**e**) as a ratio of the values after (**d**) and before (**a**) the challenge. No significant differences were found among treatments.

**Figure 9 ijms-23-06284-f009:**
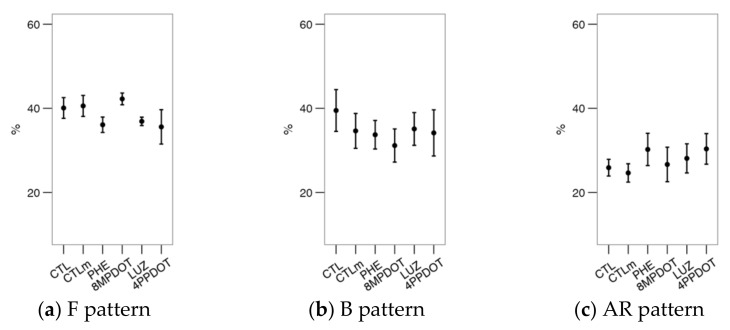
Effects on red deer epididymal sperm CTC patterns (Ca^2+^ distribution from CTC stain) of melatonin receptors agonists (PHE: Phenylmelatonin; 8MPDOT: 8-Methoxy-2-propionamidotetralin) or antagonists with 10 nM melatonin (LUZ: Luzindole; 4PPDOT: cis-4-Phenyl-2-propionamidotetralin); CTL: Control; CTLm: Control with 10 nM melatonin. Plots show mean ± SEM for each treatment of five males. (**a**) % of spermatozoa showing F pattern (non-capacitated spermatozoa showing CTC fluorescence over the head); (**b**) % of spermatozoa showing B pattern (capacitated spermatozoa showing CTC fluorescence in the acrosomal-equatorial region); (**c**) % of spermatozoa showing an AR pattern (acrosome-reacted spermatozoa showing CTC fluorescence as a band on the equatorial region). No significant differences were found among treatments.

**Figure 10 ijms-23-06284-f010:**
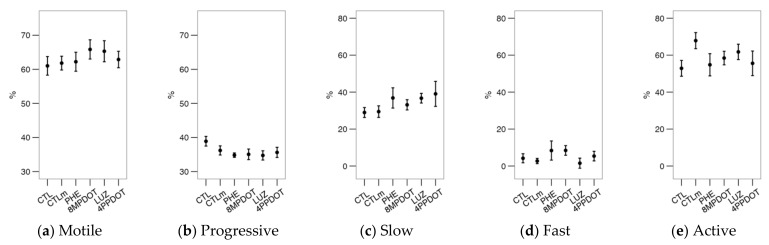
Effects on red deer epididymal sperm motility of melatonin receptors agonists (PHE: Phenylmelatonin; 8MPDOT: 8-Methoxy-2-propionamidotetralin) or antagonists with 10 nM melatonin (LUZ: Luzindole; 4PPDOT: cis-4-Phenyl-2-propionamidotetralin); CTL: Control; CTLm: Control with 10 nM melatonin. Plots show mean ± SEM for each treatment of five different males. (**a**) Percentage of motile spermatozoa; (**b**) Percentage of progressive spermatozoa; (**c**) “Slow” spermatozoa, low velocity and linearity; (**d**) “Fast” spermatozoa, high velocity and linearity; (**e**) “Active” spermatozoa, highest velocity and high lateral movement No significant differences were found among treatments.

**Table 1 ijms-23-06284-t001:** Descriptive statistics of the bull sperm subpopulations yielded by the clustering of observations from CASA data. Parameters for each subpopulation are summarized by the median ± SD of each CASA variable. The last column is the overall proportion of each subpopulation.

Subpopulation	VCL	VSL	VAP	LIN	STR	WOB	ALH	BCF	DNC	DNCm	%
Slow	59.6 ± 29.3	16.7 ± 13.5	33.8 ± 17.8	29.5 ± 18.1	54.1 ± 26.8	57.4 ± 14.0	1.6 ± 0.6	7.9 ± 5.1	95.3 ± 74.9	5.5 ± 3.9	11.6
Fast	103.4 ± 31.1	68.1 ± 24.6	78.9 ± 22.8	68.0 ± 16.0	89.3 ± 8.6	77.4 ± 11.3	1.7 ± 0.5	21.6 ± 8.0	175.1 ± 93.1	2.5 ± 1.1	22.6
Active	212.7 ± 52.1	111.5 ± 42.0	136.5 ± 28	54.9 ± 17.4	88.0 ± 11.0	64.7 ± 10.9	3.8 ± 1.5	26.0 ± 5.9	809.0 ± 495.3	7.3 ± 4.5	65.8

VCL: Curvilinear velocity; VSL: Straight path velocity; VAP: Average path velocity according to the average smoothed path; µm/s; LIN: Linearity; STR: Straightness; WOB: Wobble; ALH: Amplitude of the lateral displacement of the sperm head; BCF: Frequency of the flagellar beat; DNC: Sperm dance; DNCm: Sperm mean dance.

**Table 2 ijms-23-06284-t002:** Descriptive statistics of the red deer sperm subpopulations yielded by the clustering of observations from CASA data. Parameters for each subpopulation are summarized by the median ± SD of each CASA variable. The last column is the overall proportion of each subpopulation.

Subpopulation	VCL	VSL	VAP	LIN	STR	WOB	ALH	BCF	DNC	DNCm	%
Slow	74.5 ± 37.0	13.5 ± 10.0	34.1 ± 17.2	20.5 ± 16.4	46.6 ± 29.6	48.9 ± 15.7	1.9 ± 0.8	9.0 ± 5.9	144.1 ± 120.2	9.4 ± 8.8	27.0
Fast	126.8 ± 52.8	6.9 ± 6.7	48.0 ± 24.9	5.5 ± 5.2	14.1 ± 12.9	40.0 ± 14.8	3.4 ± 1.0	9.6 ± 5.6	439.1 ± 305.7	55.8 ± 40.1	3.0
Active	176.5 ± 60.8	82.0 ± 50.6	126.8 ± 40.9	48.8 ± 27.9	75.0 ± 26.1	68.5 ± 16.2	2.9 ± 1.1	25.6 ± 9.5	499.2 ± 342.8	6.2 ± 5	70.0

VCL: Curvilinear velocity; VSL: Straight path velocity; VAP: Average path velocity according to the average smoothed path; µm/s; LIN: Linearity; STR: Straightness; WOB: wobble; ALH: amplitude of the lateral displacement of the sperm head; BCF: Frequency of the flagellar beat; DNC: Sperm dance; DNCm: Sperm mean dance.

## Data Availability

Not applicable.

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
