# Peer review of "Bos taurus and Cervus elaphus as Non-Seasonal/Seasonal Models for the Role of Melatonin Receptors in the Spermatozoon"

_ijms, 2022, doi:10.3390/ijms23116284_

Round 1
Reviewer 1 Report
In the current study, the authors have investigated the role of melatonin receptors in the spermatozoon of Bos taurus and Cervus elaphus as non-seasonal/seasonal models. This is a straightforward study and provide some novel information on melatonin research. To improve the quality, several issue should be addressed.
- Melatonin is synthesized in the mitochondria and many of its functions may be mediated by the melatonin receptors in the mitochondria (Tan DX, Hardeland R. Targeting Host Defense System and Rescuing Compromised Mitochondria to Increase Tolerance against Pathogens by Melatonin May Impact Outcome of Deadly Virus Infection Pertinent to COVID-19. Molecules. 2020 Sep 25;25(19):4410. doi: 10.3390/molecules25194410.). This should be at least discussed in the text.
- Melatonin exhibits receptor-dependent and receptor-independent activities. In the case of deer spermatozoa, to test their stress responses such as cold, heat or chemical stress under melatonin supplementation condition will provide additional valuable information for this study.
- The authors should identify the sample sizes in each study for readers to judge the statistical analysis. This information is absent in the legends to the figures or in the statistical analysis section.
Author Response
In the current study, the authors have investigated the role of melatonin receptors in the spermatozoon of Bos taurus and Cervus elaphus as non-seasonal/seasonal models. This is a straightforward study and provide some novel information on melatonin research. To improve the quality, several issue should be addressed.
Response: The authors thank the comments and help from the reviewer.
Melatonin is synthesized in the mitochondria and many of its functions may be mediated by the melatonin receptors in the mitochondria (Tan DX, Hardeland R. Targeting Host Defense System and Rescuing Compromised Mitochondria to Increase Tolerance against Pathogens by Melatonin May Impact Outcome of Deadly Virus Infection Pertinent to COVID-19. Molecules. 2020 Sep 25;25(19):4410. doi: 10.3390/molecules25194410.). This should be at least discussed in the text.
Response: The authors appreciate the reviewer's suggestion. Consequently, we have included a piece of brief information about it in the discussion of the new version of the manuscript.
Melatonin exhibits receptor-dependent and receptor-independent activities. In the case of deer spermatozoa, to test their stress responses such as cold, heat or chemical stress under melatonin supplementation condition will provide additional valuable information for this study.
Response: The authors agree that the reviewer's suggestion would deepen the knowledge of melatonin´s physiological significance in red deer spermatozoa. This study is a first step for getting baseline knowledge about the activity of melatonin receptors on ruminants spermatozoa. Future studies will follow these methods for studying the physiological response to melatonin in sperm under stress.
The authors should identify the sample sizes in each study for readers to judge the statistical analysis. This information is absent in the legends to the figures or in the statistical analysis section.
Response: The authors thank the reviewer for their appreciation. We have included this information in the legends to the figures and also in the statistical analysis in the new version of the manuscript.
Reviewer 2 Report
Dear authors,
please find in the document attached my comments.
Congratulations for the work,
Sincerely

Author Response
The researcher Fernández-Alegre and collaborators have studied the effect of melatonin and various of its agonists/antagonists on sperm physiology during capacitation. The work covers a wide range of crucial sperm parameters and processes happening within capacitation, which proved a picture of how sperm respond to this substance, highly promising when applied in assisted reproduction techniques. Although the work could be of interest to the reader, it has serious defects that do not allow it to be accepted for publication. In general terms, the work is very difficult to understand and extremely long. The discussion needs to be reduced. Regarding the difficulty of following the thread of the work, the authors should standardize the way of naming the compounds used throughout the manuscript and avoid the use of M1-M2 agonist/antagonist. The reader cannot follow the results and the hypotheses derived from them, since when reviewing the graphs, he or she first has to know which substances are agonists or antagonists. Furthermore, it should be emphasized which substances were combined with melatonin and which were not. In fact, the results should be described respecting this aspect, comparing the treatments with the corresponding control. On the other hand, the authors must review and rewrite most of the results since there are errors in the interpretation of the graphs. Mainly, when they refer to the significance. There are many cases in which the authors highlight differences that statistically are not. Authors can find marked errors throughout the text. The authors launch hypotheses without being based on previous works and in other cases, based on erroneous interpretations of the results. Finally, after reviewing the manuscript, I am not able to distinguish whether melatonin receptors are or are not responsible of the melatonin mediated modification of spermatozoa previously observed within capacitation. In this sense, such modifications have been observed in bovine but not in deer, should not be necessary to demonstrate that melatonin modifies deer sperm physiology during capacitation?? Is it possible to promote in vitro capacitation in deer with heparin? To my mind, the general focus of the study should be modified. Due to these reasons and the defects detailed throughout the manuscript, I cannot propose the work for publication without first undergoing a major review of it.
Response: The authors thank the reviewer for their contributions to improving the manuscript. Therefore, we have made the suggested changes to the manuscript, adding specific information and restructuring sentences, for their correct interpretation by readers.
We have followed the advice for reducing the discussion and for simplifying the results. We agree that in many places results description and interpretation were mixed. Therefore, we have tried to make results clearer in this respect. We agree that the discussion was overly wordly and confusing or little clear in some aspects, therefore we have tried to improve it addressing the reviewer’s requests while shortening the text.
A limitation of the study on deer is that the samples were obtained from the epididymis, and therefore it seems refractory to capacitation stimuli in general. We have included it in the manuscript because the information provided could be helpful for future studies for this kind of sample and studies on wild ruminants.
We have also addressed the comments in the results section. We have homogenized the figures axes as requested and especially tried to clarify the explanations of the ratios (maybe more confusing due to the IRDM structure followed by IJMS and because we used ratios in two situations, as “viability” ratios and “challenge” ratios).
About the suitability of the agonists and antagonists used in this study, they have been amply validated as specific for melatonin receptors, including the different affinity. They are used in several studies cited within the manuscript with the same purpose, on spermatozoa and other cell types. We have included the catalog numbers and providers in Methods. The providers offer ample information and bibliography in their product information sheets, and offer these chemicals especifically as MT1/MT2 agonists/antagonists.
Finally, the reviewer thoughfully inquired about the suitability of melatonin for preserving deer spermatozoa, considering the few effects observed on this study. In this part of the discussion, we were referring to a previous line of work using antioxidants in order to preserve deer (and other species) spermatozoa. In that line, we used melatonin at the mM level in order to take advantage of its antioxidant effects rather than for receptor stimulation. Indeed, these concentrations were very efficient reducing the effect of oxidative effect.
Reviewer 3 Report
“Bos taurus and Cervus elaphus as non-seasonal/seasonal models for the role of melatonin receptors in the spermatozoon”
In the present study the authors aimed at analysing the roles of melatonin in the sperm physiology and functions on 2 ruminant species (domestic vs wild, non-seasonal vs very seasonal).
The work has a solid experimental design, methods utilised to analyse the sperm functions are very advanced and sophisticated, results are exciting and clearly presented and conclusions follow adequately the results that were obtained.
English language is correct, but additional revision by native English scientist should be realised.
I have just some comments/suggestions.
Abstract and Introduction
-Please revise phrase construction at Abstract (Line 14)
-On the same Line, revise spelling “hormonal” roles instead of hormone roles.
-Still in the Abstract, L 21 to L 23 revise phrase construction and clarity of the following sentence “In bull, MT1 was related to (…) sperm capacitation”.
Introduction
-Very complete but it may be useful for the reader to know where is melatonin produced in the male reproductive system (by L 42). For instance, in the anexa glans?
Results and Discussion
-Caption of the Tables could be more clarified as it is not possible to see, without going to the text, to which treatment was sperm subjected. Please clarify.
-If melatonin doesn’t affect the intracellular Ca2+, which may be the melatonin mechanism for the influence in the sperm capacitation-like state?
-Besides, if either stimulation or inhibition of the receptors decrease the AR, how melatonin acts as a player in this particular function?
-What about eventual effects of melatonin at sperm DNA integrity level?
Material and methods
In the present work, both ejaculated and epididymal sperm were used to clarify the role of melatonin. As probably expected by the authors, results were quite different as at molecular level significant differences exist between ejaculated versus epididymal sperm, besides the difference between the two species. DNA integrity was not evaluated.
Some minor points and doubts about methodology used.
-Seven bull ejaculates were collected from different breeds. Any explanation? Please clarify. Animals were young adults, still, although it should not have impact. Were there any quality criteria in order to include an ejaculate in the present study?
-Each bull’s ejaculate was split into 6 aliquots. Each ejaculate was tested just once or replicates were done with each ejaculate?
-In the red deer samples: how long after death of the animals were testicles harvested, and how long until arriving to the lab?
-On the other hand, how many red deer testicles were harvested? Please clarify.
-In the aliquots prepared, if MT1/MT2 receptors (both agonists and antagonists) were always evaluated simultaneously how was MT1 receptor evaluated individually?
-There are 2 types of controls: 0 nM and 10 nM. Please clarify the selection of 10 nM melatonin as control.
Author Response
In the present study the authors aimed at analysing the roles of melatonin in the sperm physiology and functions on 2 ruminant species (domestic vs wild, non-seasonal vs very seasonal).
The work has a solid experimental design, methods utilised to analyse the sperm functions are very advanced and sophisticated, results are exciting and clearly presented and conclusions follow adequately the results that were obtained.
English language is correct, but additional revision by native English scientist should be realised.
Response: The authors appreciate the kind comments of the reviewer.
I have just some comments/suggestions.
Abstract and Introduction
Please revise phrase construction at Abstract (Line 14). On the same Line, revise spelling “hormonal” roles instead of hormone roles.
Response: The authors thank the reviewer for their appreciation. We agree to change the sentence for the correct understanding of the reader. Thus, we have included the new sentence in lines 14 and 15 of the new version of the manuscript.
Still in the Abstract, L 21 to L 23 revise phrase construction and clarity of the following sentence “In bull, MT1 was related to (…) sperm capacitation”.
Response: The authors appreciate the reviewer's suggestion. Therefore, we have changed the phrase for its correct interpretation by the reader.
Introduction
Very complete but it may be useful for the reader to know where is melatonin produced in the male reproductive system (by L 42). For instance, in the anexa glans?
Response: The authors thank your suggestion and agree that information on the origin of melatonin in seminal plasma might be relevant to readers. Therefore, we have added a brief information about it in the new version of the manuscript (lines 45-47).
Results and Discussion
Caption of the Tables could be more clarified as it is not possible to see, without going to the text, to which treatment was sperm subjected. Please clarify.
Response: Thank you for indicating that (we believe that the reviewer refers to the Figure captions). We have added the treatment definitions to the captions.
If melatonin doesn’t affect the intracellular Ca2+, which may be the melatonin mechanism for the influence in the sperm capacitation-like state?
Response: Many pathways modulate the capacitation. Melatonin might be affecting cAMP synthesis or degradation, which was suggested previously. We have mentioned this in the discussion, but as the reviewer comments, it could be important to highlight it. We have modified part of the discussion to better clarify it.
Besides, if either stimulation or inhibition of the receptors decrease the AR, how melatonin acts as a player in this particular function?
Response: The use of antagonists increased AR in bull (Fig. 1d and 3a), especially for 4PPDOT and when comparing with the CTL+mel, with no effects of agonists or antagonists for deer. This is true that when using the ionophore or LPC, there were no differences between these treatments. We have revised and clarified the description of these results.
What about eventual effects of melatonin at sperm DNA integrity level?
Response: We have not considered DNA integrity in this study, since our previous studies showed that melatonin protected the sperm DNA at relatively high concentrations (millimolar) and when the cells were submitted to high oxidative stress. Moreover, DNA damage is less evident in ruminants than in humans, for instance, at least when the samples are obtained from good quality males. Therefore, we focused on those parameters potentially more relevant for determining the effects of melatonin receptors.
Material and methods
In the present work, both ejaculated and epididymal sperm were used to clarify the role of melatonin. As probably expected by the authors, results were quite different as at molecular level significant differences exist between ejaculated versus epididymal sperm, besides the difference between the two species. DNA integrity was not evaluated.
Response: The reviewer is right. For future studies, comparing epididymal and ejaculate spermatozoa in the bull model could clarify this. Such a study could be carried out in deer too, however more complex due to the obtention of ejaculates (electroejaculation).
Some minor points and doubts about methodology used.
Seven bull ejaculates were collected from different breeds. Any explanation? Please clarify. Animals were young adults, still, although it should not have impact. Were there any quality criteria in order to include an ejaculate in the present study?
Response: The bull samples were obtained in collaboration with the Animal Selection and Reproduction Center - CENSYRA. In their workflow, they take samples from different breeds of bull and we thought it would be interesting for the experiment to evaluate them and rule out possible differences between males or breeds. As explained in the material and methods section, the ages of the animals housed there range between 12 and 14 months. They are males with good genetic qualities and great seminal quality since they pass the CENSYRA criteria to be cryopreserved for AI and germplasm preservation.
Each bull’s ejaculate was split into 6 aliquots. Each ejaculate was tested just once or replicates were done with each ejaculate?
Response: Thanks for your appreciation. Effectively, each bull's ejaculate was divided into 6 tubes, which were incubated under the conditions described in the manuscript. After incubation, duplicate samples were taken from each of the tubes for each analysis. We have added this information in the new version of the manuscript.
In the red deer samples: how long after death of the animals were testicles harvested, and how long until arriving to the lab? On the other hand, how many red deer testicles were harvested? Please clarify.
Response: The authors agree that the information requested by the reviewer is crucial. We have included the number of males and the time elapsed from the hunting of the animal to its analysis in the laboratory in the section Reagents, animals, and sample collection and preparation of the material and methods. As we have confirmed previously, even if the testicles are quickly refrigerated, there is a negative effect of the time on sperm quality. Therefore, the testicles were kept within the scrotal sac (a very effective measure as confirmed by us), and quickly sent refrigerated to our lab, processing them as soon as the arrived on the following day.
In the aliquots prepared, if MT1/MT2 receptors (both agonists and antagonists) were always evaluated simultaneously how was MT1 receptor evaluated individually?
Response: This question is very relevant to our study, and this fact driven our experimental design (and previous studies using a similar one in other species). As far as we know, there are no specific agonist/antagonists for the MT1 receptor. In fact, 8M-PDOT and 4P-PDOT are not entirely specific for MT2. Nevertheless, 8M-PDOT and 4P-PDOT show a much higher affinity for MT2 receptor. Therefore, by using appropriate concentrations, the differences found between phenylmelatonin (equal affinity for both receptors) and 8M-PDOT (higher affinity for MT2) or between luzindole (equal affinity for both receptors) and 4P-PDOT (higher affinity for MT2) allow to determine (with varying degrees of certainty depending on effect sizes) the participation of each receptor in the responses.
There are 2 types of controls: 0 nM and 10 nM. Please clarify the selection of 10 nM melatonin as control.
Response: We have added information to clarify the use of two controls in the new version of the manuscript.